# Bodybuilding Coaching Strategies Meet Evidence-Based Recommendations: A Qualitative Approach

**DOI:** 10.3390/jfmk8020084

**Published:** 2023-06-16

**Authors:** Alexa Rukstela, Kworweinski Lafontant, Eric Helms, Guillermo Escalante, Kara Phillips, Bill I. Campbell

**Affiliations:** 1Performance and Physique Enhancement Laboratory, Exercise Science Program, University of South Florida, Tampa, FL 33620, USA; arukstela@usf.edu (A.R.); kworweinski@usf.edu (K.L.); kara.phillips@unco.edu (K.P.); 2Sports Performance Research Institute New Zealand, Auckland University of Technology, Auckland 1010, New Zealand; eric.helms@aut.ac.nz; 3Muscle Physiology Laboratory, Department of Exercise Science and Health Promotion, Florida Atlantic University, Boca Raton, FL 33431, USA; 4Department of Kinesiology, California State University, San Bernardino, CA 92407, USA; gescalan@csusb.edu; 5Department of Kinesiology, Nutrition, and Dietetics, University of Northern Colorado, Greeley, CO 80639, USA

**Keywords:** fat loss, supplements, performance-enhancing drugs, cardiovascular exercise, programming

## Abstract

Bodybuilding is a sport where coaches commonly recommend a variety of nutrition and exercise protocols, supplements, and, sometimes, performance-enhancing drugs (PEDs). The present study sought to gain an understanding of the common decisions and rationales employed by bodybuilding coaches. Focusing on coaches of the more muscular divisions in the National Physique Committee/IFBB Professional League federations (men’s classic physique, men’s bodybuilding, women’s physique, women’s bodybuilding) for both natural and enhanced athletes, coaches were recruited via word of mouth and social media, and 33 responded to an anonymous online survey. Survey responses indicated that participant coaches recommend three-to-seven meals per day and no less than 2 g/kg/day of protein regardless of sex, division, or PED usage. During contest preparation, participant coaches alter a natural competitor’s protein intake by −25% to +10% and an enhanced competitor’s protein intake by 0% to +25%. Regarding cardiovascular exercise protocols, approximately two-thirds of participant coaches recommend fasted cardiovascular exercise, with the common rationale of combining the exercise with thermogenic supplements while considering the athlete’s preference. Low- and moderate-intensity steady state were the most commonly recommended types of cardiovascular exercise among participant coaches; high-intensity interval training was the least popular. Creatine was ranked in the top two supplements for all surveyed categories. Regarding PEDs, testosterone, growth hormone, and methenolone were consistently ranked in the top five recommended PEDs by participant coaches. The results of this study provide insight into common themes in the decisions made by bodybuilding coaches, and highlight areas in which more research is needed to empirically support those decisions.

## 1. Introduction

Bodybuilding is a sport where athletes are judged on the aesthetic qualities of their physique such as muscle size, low body fat, and overall symmetry. To achieve this look, bodybuilders typically train for years to increase muscle mass, then enter a preparation/dieting phase, with the aim of reducing body fat to extremely low levels while maintaining high levels of muscularity. The previous literature investigating the varying nutrition strategies [1], supplementation, and training practices [2] over the competitive seasons of individual bodybuilders reported a blend of evidence-based practice and unsubstantiated anecdotal practice.

These previously mentioned survey studies on bodybuilding investigated from the perspective of the athlete themselves; to the authors’ knowledge, little attention has been given to the bodybuilding coaches who often orchestrate these competition preparations. Given that many bodybuilders hire coaches, the purpose of this investigation was to identify the common coaching practices employed by coaches working with athletes that compete in the largely non-drug-tested IFBB Professional League or the National Physique Committee (NPC). Additionally, we sought to examine the tactics applied to the most muscular divisions in men’s and women’s bodybuilding by specifically analyzing supplementation and performance-enhancing drug (PED) usage. While all bodybuilding classes require a foundation of muscle, the select divisions (women’s bodybuilding, women’s physique, men’s bodybuilding, and men’s classic physique) place the largest emphasis on overall muscle mass. Therefore, our investigation included only coaches for athletes competing in those divisions.

To the authors’ knowledge, the present study is the first investigation into the strategies used by bodybuilding coaches. This research will be of interest to bodybuilding coaches and athletes looking to gain a deeper understanding of common practices in their sport. The current study will also be of interest to researchers seeking to investigate or understand the methods used to build some of the most muscular physiques in a sport that rewards such outcomes. Finally, this work is novel in its discussion of the preferred PEDs of coaches recommending these compounds for enhanced, rather than drug-tested “natural”, bodybuilding athletes.

## 2. Materials and Methods

### 2.1. Participants

Bodybuilding coaches were recruited through social media advertisements and word of mouth. All participant coaches were required to coach athletes that compete in the IFBB Professional League/NPC and were required to have coached any of the two most muscular categories for male and female bodybuilders including men’s bodybuilding, men’s classic physique, women’s bodybuilding, and women’s physique. Participant coaches were required to have coached someone in one of these categories in the past five years within the IFBB Professional League/NPC and not respond to any survey questions in reference to their athletes competing in other divisions such as bikini, wellness, figure, or men’s physique. This observational study was considered “minimal risk” and approved by the University of South Florida Institutional Review Board (IRB ID: STUDY004054). This research was carried out fully in accordance with the ethical standards of the Declaration of Helsinki, and informed consent was provided by participants prior to beginning the survey.

### 2.2. Protocol

The anonymous online survey was developed collaboratively by the authors, several of whom are bodybuilding competitors and coaches, and conducted using Qualtrics software (Qualtrics, Provo, UT, USA). Prior to distribution, the survey was tested by volunteers to ensure the software operated as intended. The survey was circulated internationally in English via social media (Instagram, Twitter, and Facebook) and e-mail, with data collected between 21 April 2022 and 16 June 2022. Participants were asked 41 questions regarding their bodybuilding coaching practices. These questions were categorized into 5 major topics: population (5 questions), protein intake (10 questions), cardiovascular exercise (12 questions), supplementation (8 questions), and PEDs (6 questions). These questions sought to identify common coaching practices in these domains to compare them to current evidence-based recommendations, as well as compare the differences between natural and enhanced competitors, and male and female competitors. Participant coaches were allowed to skip any question they did not want to answer or feel qualified to answer based on their background and experiences, which occurred on an average of 3.6 questions per survey participant. The survey was designed to only display questions to coaches which were in alignment with the demographics (i.e., male/female; natural/enhanced athletes) that they indicated having coaching experience in at the beginning of the survey.

### 2.3. Statistical Analysis

Frequencies were calculated on all data to determine and rank responses by mode, and descriptive statistics were calculated via Microsoft Excel for numerical data to report ranges and means. Qualitative data were reported as-is with no attempt to further analyze or extrapolate, including direct quotes and paraphrased responses. The analysis of the data remained divided by respective topic area.

## 3. Results

### 3.1. Population

A total of 33 bodybuilding coaches completed the online survey. Participants coached athletes from regional NPC to top level IFBB Professional League Olympia and Arnold Classic contests. Coaching résumés included ranges for the following milestones: class wins: 1–72; overall wins: 0–44; top five national placings: 0–30; professional card winners: 0–37; professional show wins: 0–10; range of athletes competing at the highest level in the Arnold Classic or Olympia: 0–2. Data were not collected on further demographics of sex, race, or socioeconomic status.

### 3.2. Protein

Participant coaches reported recommending their natural athletes (who do not use PEDs) between 2–3.3 g of protein/kg bodyweight (0.9–1.5 g/lb) per day in the off-season. Enhanced (PED-using) female competitors were recommended between 2–3.85 g/kg of bodyweight (0.9–1.75 g/lb) of protein per day in the off-season. Enhanced male competitors were recommended between 2–4.84 g/kg bodyweight (0.9–2.2 g/lb) of protein per day in the off-season. Meal frequency was between 3–7 meals per day. The percentage of change of protein intake during the preparatory season ranged from an increase of 10% to a decrease of 25% in natural athletes. For enhanced athletes, the changes in protein intake during the preparatory season ranged from no change to an increase of 25%.

### 3.3. Cardiovascular Exercise

Approximately 67% of participant coaches reported utilizing fasted cardiovascular exercise with their athletes. Common reasons for using fasted cardiovascular exercise included the concurrent use of fat-burning supplements as well as the athlete’s preferred time of day to complete cardiovascular exercise. Cardiovascular exercise was most commonly tracked as number of minutes per session, but was also monitored through step count, heart rate, and calories expended. Commonly recommended cardiovascular exercise modalities included treadmill, stair-master, elliptical, and stationary bike, with the treadmill as the most prevalent modality (Table 1). Relative to exercise intensity, low-intensity steady state (LISS) and moderate-intensity steady state (MISS) were the most common across all groups. High-intensity interval training (HIIT) was almost never mentioned, with the only exception being in the enhanced male competitor category. On a weekly basis, cardiovascular exercise duration was higher in female competitors, ranging from 40 to 740 min per week for both natural and enhanced female competitors. Cardiovascular exercise duration was between 0 and 480 min per week for enhanced male competitors, and 40 and 480 min per week for natural male competitors.

### 3.4. Supplementation

The top five most frequently recommended supplements are included in Table 2, and are listed by mode with ties included. Creatine was the number one recommended supplement across all categories except for enhanced male competitors in the off-season, where it placed second. Fish oils/omega-3 fatty acids were also included in all top five lists. Protein was in the top five most commonly recommended supplements for all categories except male enhanced competitors in the off-season. In preparatory phases, caffeine was utilized for all categories. In male athletes, both natural and enhanced, yohimbine was recommended during the preparatory season. Ashwagandha was recommended to female enhanced athletes during both the preparatory phase and the off-season, as well as male natural athletes during preparation. Pre-workout, defined as a blend of ingredients designed to elicit an acute response favorable to exercise performance [3], was commonly recommended to natural athletes during preparation, as well as enhanced female athletes in the off-season. Male enhanced athletes were also commonly recommended bergamot.

### 3.5. Performance-Enhancing Drugs

The most commonly recommended PEDs are listed in Table 3. PEDs commonly recommended to female competitors included testosterone, clenbuterol, oxandrolone (anavar), methenolone (primobolan), and growth hormone. The most commonly recommended PEDs for male competitors included testosterone, methenolone enanthate (Primobolan), growth hormone, drostanolone propionate (masteron), insulin, and decadurabolin (trenbolone). Dosages expected for lower-level enhanced male athletes ranged between 600 and 1500 mg/week of total substances. Dosages expected for higher level enhanced male athletes ranged between 300 and 4000 mg/week. Dosages for lower-level enhanced female athletes ranged between 5 and 150 mg/week of total substances, while higher level enhanced female athletes could be expected to range between 30 and 200 mg/week.

## 4. Discussion

### 4.1. Protein

Participant coaches reported recommending daily protein intakes ranging from 2.0 to 4.84 g of protein per kilogram of body mass. Existing evidence currently supports the ingestion of 1.4–2.0 g/kg/day of protein for physically active individuals [4], meaning participant coaches often recommend 2–3x the recommended protein levels found in the literature. Levels above this amount have not been shown to provide significant increases in lean mass, as best exemplified in a recent meta-analysis by Nunes et al. where additional benefits began to diminish above protein intakes of 1.6 g/kg/day [5]. While the populations included in the position stand led by Jäger et al. were often trained, they were not necessarily highly muscular competitive bodybuilders [4]. Based on the responses of participant coaches, it could be possible that natural and enhanced bodybuilders with more muscle mass than average gym-goers may require higher protein intakes. At present, multiple recent systematic reviews with meta-analyses report that protein intakes of around 1.6 g/kg/day or higher, on average, appear to maximize gains in lean mass [5,6], although Morton and colleagues also recommended 2.2 g/kg/day “for those seeking to maximize resistance training-induced gains in FFM” as a pragmatic guideline, as this was the upper end of their 95% confidence interval [6]. Therefore, additional research, particularly in highly muscular, enhanced, and natural competitive bodybuilders, is needed as this population is currently understudied in relation to optimal protein intakes.

A protein-feeding frequency of 3–7 meals per day was reported by the participant coaches, which is in alignment with the current evidence-based recommendations for the bodybuilding population of between 3 and 6 protein feedings per day [7]. In addition to protein-feeding frequency, recent research has also highlighted that the amount of protein intake per feeding should be approximately evenly distributed throughout the day to theoretically optimize lean mass accretion, with total protein absorption reaching a limit at approximately 0.60 g/kg/meal for older men and 0.40 g/kg/meal for younger men [8,9]. The present authors are unaware of any similar research on protein absorption per meal in women. Further, these speculations are based on mechanistic data. There are no studies showing superior muscle mass accretion in well-trained lifters due to more even protein distribution of protein across higher meal frequencies in comparable ranges to those reported in this survey versus lower frequencies or uneven distributions, and thus, more research is needed in this area as well.

Participant coaches reported increasing protein intake by up to 25% during contest preparation in enhanced athletes, while changing protein intake by between +10% and –25% in natural athletes. The current recommendation is to maintain higher intakes in resistance-trained subjects during dieting phases to maximize the retention of lean mass [4]. While speculative, enhanced athletes may be able to increase protein throughout contest preparation periods as they utilize anabolic androgenic steroids which aid with the retention, and potentially, continued growth of fat-free mass. Enhanced athletes are also likely to have more options to stimulate fat loss with the utilization of compounds such as growth hormone, clenbuterol, and triiodothyronine (T3) which would reduce the necessity of decreasing protein to create a larger deficit. Given that natural athletes abstain from these compounds, some coaches may consider decreasing absolute protein amounts to create the deficit required to achieve stage leanness. However, this reduction should be modest as several groups of authors have postulated that higher protein intakes during energy restriction may help reduce the loss of lean body mass in resistance-trained athletes [10,11,12].

### 4.2. Cardiovascular Exercise

Fasted cardiovascular exercise was recommended by approximately two-thirds of participant coaches, with the most common reason for preference being the added utilization of yohimbine, growth hormone, and clenbuterol (their combination with fasted exercise is thought to enhance fat loss). A systematic review reported that fasted cardiovascular exercise is not more effective than fed cardiovascular exercise [13]. Notably, none of the studies included in the systematic review recruited competitive bodybuilders as subjects, nor did they look at subjects utilizing fasted cardiovascular exercise in tandem with these specific PEDs for an additional fat-burning effect. A review article by Escalante et al. suggests that physique athletes may perform fasted cardio at varying intensities, but it is not suggested for longer than 60 min to prevent fat-free mass losses [14]. Future research in this population is needed in which fasted cardiovascular exercise is paired with fat-burning compounds to determine if this pairing is (1) safe and (2) efficacious for fat loss outcomes.

Participant coaches notably did not prefer HIIT cardiovascular exercise over LISS or MISS cardiovascular exercise for any of the populations. This recommendation is in line with previous research, as a systematic review and meta-analysis reported similarly effective fat-burning effects between HIIT and MISS [15]. Furthermore, authors of a narrative review suggested that HIIT cardiovascular exercise should not be used too frequently due to the increased recovery demands during natural bodybuilding contest preparation [16]. Further research in both enhanced and natural populations is warranted on this topic.

Participant coaches recommended that female athletes perform more cardiovascular exercise during contest preparation, up to 740 min (12+ h) per week, and male bodybuilding athletes up to 480 min (8 h) per week. Given the potential of high volumes of resistance training to induce additional fatigue while dieting, authors of some reviews recommend the lowest amount of cardiovascular exercise needed to achieve the desired result to mitigate any negative impact of cardiovascular training [16,17]. The range of cardiovascular exercise frequencies and durations for athletes is notably very wide, and this is likely due to the individuality of every athlete’s energy expenditure, current body composition, genetics, timeline to achieve stage-leanness, required leanness for their respective division, and individual preference. Neither coaches nor researchers may be able to provide generalizations about the amount of cardiovascular exercise that will be required to obtain the desired leanness for specific divisions due to the individuality of each athlete.

### 4.3. Supplementation

Creatine was the most commonly reported supplement across all groups. These findings are similar to previously published research where 84.4% of surveyed natural competitors and 52.5% of surveyed enhanced competitors reported ingesting creatine monohydrate (CM) [2]. Given the abundance of research on the efficacy of CM to improve strength and lean mass gains, this outcome is in alignment with the scientific evidence [18]. While CM’s performance-enhancing effects and ability to increase lean body mass are well understood, the effects of utilizing CM with PEDs is unknown and warrants future study.

Caffeine was reported as one of the most commonly recommended supplements across all groups. This is slightly in contrast to the previously cited survey report of male natural and enhanced bodybuilders, where natural bodybuilders were significantly more likely to utilize caffeine as a supplement than enhanced athletes [2]. Given caffeine’s ability to increase exercise energy expenditure [19], this recommendation is in alignment with the current scientific evidence.

Omega-3 supplementation was also commonly recommended by the participant coaches. Omega-3 supplementation can improve endurance capacity and promote recovery, but there is limited evidence to support that omega-3 fatty acid supplementation significantly contributes to muscle hypertrophy or body fat loss [20,21]. Thus, while not for bodybuilding performance specifically, omega-3 supplementation may be recommended by coaches as a general health supplement. In a systematic review and meta-analysis, both eicosapentaenoic acid and docosahexaenoic acid significantly reduced triglyceride levels when taken separately and together [22]. This effect may be of use to enhanced bodybuilders, specifically due to the strain PEDs can place on blood lipids [23].

Another supplement which made the top five recommendations of all bodybuilding coaches was bergamot for enhanced competitors, but this was not reported in natural competitors. In a systematic review, bergamot supplementation decreased total cholesterol, triglyceride levels, and LDLc [24]. When combined with omega-3 supplementation for enhanced bodybuilders, bergamot may protect against the negative blood lipid effect associated with PED use [23]. Future research in bodybuilding athletes utilizing PEDs is needed to identify supplementation strategies to offset the negative side effects of drug use.

### 4.4. PEDs

Across both male and female competitive athletes, testosterone was recommended most frequently. Unlike most anabolic androgenic steroids, testosterone has been extensively studied in healthy populations such as men undergoing testosterone replacement therapy (TRT) [25]. For men, TRT dosages typically start between 75 and 100 mg/week but can increase to 200 mg/week depending on the compound [25]. Participant coaches speculated male bodybuilders would take between 300 and 4000 mg/week of total PEDs (not testosterone alone). These values do align, however, with previous survey responses of male bodybuilding athletes who reported utilizing 400–600 mg/week of testosterone [26]. Research is limited regarding the effects of total androgenic loads of this magnitude in healthy, physically active male athletes.

Primobolan, otherwise known as methenolone oenanthate, is a PED which was also frequently reported for both male and female bodybuilders by participant coaches. In the current literature, primobolan is most often studied in those with breast cancer, anemia, and rheumatoid arthritis [27,28]. For breast cancer patients, side effects included hoarseness, weight gain, acne, hirsutism, and increases in well-being [27]. For those with rheumatoid arthritis, West et al. demonstrated that treatment with primobolan resulted in an increase in weight with an increase in back skinfold thickness, but a decrease in arm skinfold thickness; the only additional reported side effect was an increase in acne [28]. Both of these studies were performed in female patients; however, the impact of this compound on healthy female and male competitors is yet to be illuminated.

Growth hormone was also frequently recommended by participant coaches to both male and female competitors. Naturally occurring growth hormone possesses both lipolytic and anabolic effects including organ, bone, and muscle growth. This compound has been studied in healthy male and female athletes where researchers reported “growth hormone significantly reduced fat mass, increased lean body mass through an increase in extracellular water, and increased body cell mass in men when co-administered with testosterone” [29]. Another study in healthy young men noted growth hormone does not seem to provide an anabolic stimulus or increase in muscle mass on its own [30]. Finally, a systematic review of growth hormone on athletic performance stated that, while there are acute increases in lean body mass due to fluid retention, there is no improvement in strength and potentially a decrement to exercise capacity [31]. Given participant coaches frequently mentioned utilizing growth hormone synergistically with fasted cardiovascular exercise, it seems they believe there is a potential added benefit of using growth hormone in bodybuilders during competition prep. To the authors’ knowledge, this situation has not been explored in the scientific literature.

Clenbuterol was reported as one of the most common PEDs for female athletes but was suggested less frequently for male athletes. Clenbuterol is a beta-2 agonist, resulting in fat loss and increased skeletal muscle recovery. When used in healthy male subjects, there was an increase of fat oxidation by 39%, an increase in resting metabolic rate by 21%, and an increase in mTOR phosphorylation by 121% [32]. In another study of young men recovering from a meniscectomy, those in the clenbuterol group experienced quicker recovery in the quadriceps compared to the placebo group [33]. Its usage in bodybuilding has been reviewed, and researchers have advised against the recreational usage of clenbuterol due to adverse reactions such as tachycardia, widened pulse pressure, tachypnea, hypokalemia, hyperglycemia, increases in troponin and creatine phosphokinase, and tremors [34]. Overall, participant coaches seemed to recommend clenbuterol in female competitors more than male competitors, and believed there may be an additive effect of clenbuterol with fasted cardiovascular exercise; however, there is no current scientific validation of this practice.

Overall, the compounds reported by participant coaches only partially aligned with a previous survey research study in bodybuilding athletes. The five most commonly reported PEDs by male athletes in the aforementioned study were “dianabol, deca durabolin, anavar, testosterone, [and] anadrol” [35]. Of these PEDs, participant coaches in the present study most commonly included testosterone in their top five rankings, whereas anavar and anadrol were sparingly mentioned and decadurabolin and dianabol were not recommended. The female athletes’ top five reported compounds were “deca durabolin, anavar, testosterone, dianabol, and equipoise” [35]. Of these PEDs, participant coaches in the present study only frequently mentioned testosterone and anavar, with no mention of the other compounds. The reasons behind the differences between the present study and Tricker et al.’s survey [35] cannot be definitively known due to an absence of research regarding PED usage among bodybuilders, although there does appear to be a shift in compound selection. Such a shift could be due to a difference in preference between coaches and athletes, a change in commercial availability of PEDs, and/or an alteration in understanding of PEDs and their efficacy over time. When Tricker and colleagues conducted their survey of bodybuilding athletes, “controversy still [existed] as to whether steroids do, indeed, actually improve muscular size and strength,” [35] around which little to no controversy currently exists.

Currently, PED usage is illegal and banned in most sports, which also makes applied research regarding these compounds nearly impossible due to IRB concerns and leaves survey data on PEDs relatively sparse. However, the NPC/IFBB Professional League are viewed in bodybuilding circles as untested organizations which tacitly allow PED use [36]; notably, the rules overviews on the organization’s websites provide no information on anti-doping, at the time of writing. Because of this, athletes utilize these compounds and still participate in their chosen sport at their own risk. In a recent literature review, researchers evaluated the prevalence of sudden cardiac death in bodybuilding athletes and reported 33 deaths of athletes averaging 30 years old primarily due to PED misuse [23]. PED implementation is a common choice among bodybuilders, but most of the literature warns against their undesirable effects. There is a lack of research identifying safer practices for athletes and coaches choosing to implement PEDs; thus, current practices may be largely influenced by anecdotes and previous experience alone. This lack of research is likely due to the illegal nature of many PEDs and the ethical dilemma of qualified physicians and their choice in prescribing PEDs or advising coaches (who are potentially unqualified) against recommending PEDs.

### 4.5. Limitations

As this is the first study identifying common bodybuilding coaching practices, there were several limitations. The major limitation of the present study was attempting to synthesize what is currently recommended to individual bodybuilders. With the understanding that bodybuilding coaching requires an individualized approach, the present study sought to report general trends to further identify gaps in current research in bodybuilding populations. To give coaches the ability to answer freely and express the nuance of their coaching strategies, all survey questions were designed to be open-ended and optional. Definitions of what constitutes a lower-level versus upper-level athlete were also left to the participant coaches’ discretion, as individual perspectives on distinguishing criteria would be critical in answering questions. The open nature of the questions subsequently limited the standardization of the participant coaches’ answers. Upon reviewing the answers to the questions, this challenge was presented occasionally as misreported units, vague answers, or unanswered survey questions.

Another limitation of this study is the willingness of participants to complete the study. Feedback received from one reluctant participant involved the concern that the present study would “put bodybuilding coaching into boxes”, despite the extensive efforts of the researchers to create open-forum questions. Potential participants may have felt they were unable to synthesize their coaching practices, and thus were unwilling to complete our survey. The 41-item questionnaire was developed for the present study, and thus could be improved upon for any potential future use to address the concerns of the reluctant participants.

## 5. Conclusions

No two participant coaches provided identical answers, demonstrating the unique nature of coaching in the studied bodybuilding divisions. Varying the approach depending on the athlete was a common theme among participant coaches. Comparing their responses to recommendations from the scientific literature, although there were exceptions, the present study concludes that bodybuilding coaches’ decisions somewhat align with current evidence-based recommendations. Most variations from this may be a result of coaching towards individual variance between athletes, or speculations due to the lack of data. Thus, despite the clear need for further research among bodybuilding populations, current and future coaches may find it beneficial to be familiar with current evidence-based recommendations. Finally, researchers may seek to investigate the currently under-supported practices utilized by participant coaches for future study designs.

## Figures and Tables

**Table 1 jfmk-08-00084-t001:** Participant coaches cardiovascular exercise responses.

**Do you utilize fasted cardio with your athletes during preparation?**		**How do you track your athletes’ cardio? Select all that apply.**	
*Response*	*Times Mentioned*	*Response*	*Times Mentioned*
Yes	10	Minutes	12
No	5	Steps	9
		Calories/kcals	8
		Heart Rate	6
		Smart Watches	1
		Qualitative Description	1
**What mode of cardio do you specify for your athletes? Select all that apply.**		**When would you chose fasted over fed cardio?**	
*Response*	*Times Mentioned*	*Response*	*Times Mentioned*
Treadmill	11	When using a fat burner	5
Stair Stepper	10	Athlete’s Choice/Preference	4
Elliptical	8	When closer to show	1
Stationary Bike	8	When cardio volume reaches a certain point (above 60 min or 1 session a day)	1
Athlete’s Choice	5		
Ropes	1		
Rower	0		
**What type of cardio do you prefer to use for your female enhanced competitors?**		**What type of cardio do you prefer to use for your male enhanced competitors?**	
*Response*	*Times Mentioned*	*Response*	*Times Mentioned*
MISS ^2^	4	LISS ^1^	6
LISS ^1^	3	MISS ^2^	2
LISS + HIIT ^1,3^	1	HIIT ^3^	1
LISS + MISS + HIIT ^1,2,3^	1	LISS + HIIT ^1,3^	1
Athlete’s Preference	1	LISS + MISS + HIIT ^1,2,3^	1
HIIT ^3^	0	Athlete’s Preference	1
**What type of cardio do you prefer to use for your female natural competitors?**		**What type of cardio do you prefer to use for your male natural competitors?**	
*Response*	*Times Mentioned*	*Response*	*Times Mentioned*
LISS ^1^	6	LISS ^1^	6
MISS ^2^	3	MISS ^2^	3
LISS + HIIT ^1,3^	1	LISS + HIIT ^1,3^	1
LISS + MISS + HIIT ^1,2,3^	1	Athlete’s Preference	1
Athlete’s Preference	1	LISS + MISS + HIIT ^1,2,3^	0
HIIT ^3^	0	HIIT ^3^	0

^1^ LISS = Low-Intensity Steady State; ^2^ MISS = Moderate-Intensity Steady State; ^3^ HIIT = High-Intensity Interval Training.

**Table 2 jfmk-08-00084-t002:** Top recommended supplements by participant coaches.

**Male Natural** **Off-Season**			**Female Natural** **Off-Season**		
*Supplement Name*	*Times Mentioned*	*Dosage Range*	*Supplement Name*	*Times Mentioned*	*Dosage Range*
Creatine	10	3–10 g/day	Creatine	11	3–10 g/day
Protein	5	0.5 g/kg/day	Protein	5	0.5–2.5 g/kg/day
Fish Oil/Omega 3	6	2–6 g/day	Caffeine	4	200–300 mg/day
EAA ^1^	4	5–10 g/day	EAA ^1^	4	3–10 g/day
Vitamin D	4	2000–20,000 IU/day	Fish Oil/Omega 3	4	2–4 g/day
			Citrulline	4	6–7 g/day
**Male Enhanced** **Off-Season**			**Female Enhanced** **Off-Season**		
*Supplement Name*	*Times Mentioned*	*Dosage Range*	*Supplement Name*	*Times Mentioned*	*Dosage Range*
Fish Oil/Omega 3	6	0.9–5 g/day	Creatine	6	5 g/day
Creatine	5	5–10 g/day	Fish Oil/Omega 3	5	2–4 g/day
NAC ^2^	4	1.2–2.4 g/day	Protein	3	0.5 g/kg/day
Bergamot	4	0.5–2 g/day	Pre-Workout	2	–
Multi-Vitamin	3	–	Caffeine	2	200–300 mg/day
			Ashwagandha	2	500 mg/day
			Curcumin	2	400–800 mg/day
			Vitamin D	2	2000–10,000 IU/day
			Citrulline Malate	2	6 g/day
			Bergamot	2	500–1000 mg/day
			Multi-Vitamin	2	–
**Male Natural** **Preparation**			**Female Natural** **Preparation**		
*Supplement Name*	*Times Mentioned*	*Dosage Range*	*Supplement Name*	*Times Mentioned*	*Dosage Range*
Creatine	7	5–10 g/day	Creatine	8	5–10 g/day
Caffeine	5	200–400 mg/day	Caffeine	5	200–400 g/day
Protein	5	0.5 g/kg/day	Protein	5	0.5–2.5 g/kg/day
Fish Oil/Omega 3	5	2–6 g/day	Fish Oil/Omega 3	4	2–4 g/day
Yohimbine	3	5–20 mg/day	Pre-Workout	3	–
Ashwagandha	3	600 mg/day			
Pre-Workout	3	–			
**Male Enhanced** **Preparation**			**Female Enhanced** **Preparation**		
*Supplement Name*	*Times Mentioned*	*Dosage Range*	*Supplement Name*	*Times Mentioned*	*Dosage Range*
Creatine	5	5–10 g/day	Creatine	6	5 g/day
Fish Oil/Omega 3	5	2–5 g/day	Fish Oil/Omega 3	5	2–6 g/day
Protein	4	0.5 g/kg/day	Protein	3	0.5 g/kg/day
Yohimbine	4	5–20 mg/day	Caffeine	3	200–300 mg/day
Caffeine	3	200–300 mg/day	Ashwagandha	3	600–1000 mg/day
Bergamot	3	0.5–2 g/day			

^1^ EAA = Essential Amino Acids; ^2^ NAC = N-Acetyl Cysteine; “–“ = No dosage range reported.

**Table 3 jfmk-08-00084-t003:** Reported PEDs recommended by participant coaches for enhanced athletes.

Male Competitors		Female Competitors	
*PED Name*	*Times Mentioned*	*PED Name*	*Times Mentioned*
Testosterone	15	Testosterone	10
Primobolan	11	Clenbuterol	10
Growth Hormone	9	Anavar	9
Masteron	8	Primobolan	9
Insulin	6	Growth Hormone	8
Trenbolone	6	Insulin	5
Clenbuterol	3	T3 ^1^	4
Nandrolone Phenylpropionate	3	Metformin	2
Anavar	2	Winstrol	2
Winstrol	2	Proviron	2
Proviron	1	Methenolone	1
Methenolone	1	Masteron	1
Anadrol	1	Nandrolone Phenylpropionate	1
Aromasin	1	T3/T4 ^1,2^	1
Metformin	1	T4 ^2^	1
Equipoise	1	Turinabol	1
		Injectable L-Carnitine	1
		Telmisartan	1

^1^ T3 = Triiodothyronine; ^2^ T4 = Thyroxine.

## Data Availability

The data presented in this study are available in the Results section of the present article, Section 3.1, Section 3.2, Section 3.3, Section 3.4 and Section 3.5.

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
