# Peer review of "Bodybuilding Coaching Strategies Meet Evidence-Based Recommendations: A Qualitative Approach"

_jfmk, 2023, doi:10.3390/jfmk8020084_

Round 1
Reviewer 1 Report
Thank you for your efforts on this submission. The research question included novelty in targeting coaches rather than athletes and the unique aspect of PEDs. The writing and presentation throughout were direct and mostly clear.
Introduction
Set up of the question for the reader was clear.
Materials and Methods
Mostly clear with one minor and one more impactful question:
Minor: Regarding social media distribution, if only a few platforms were used, could they be listed? If many, I do not believe including a list would be worthwhile.
More impactful: Was there any strategy employed to account for coaches of athletes fitting into multiple categories (male/female and/or natural/PED-using competitors)? Would/Could these coaches fill out the survey multiple times, say once answering regarding the training of the male clients and once regarding the training of their female clients? If not, how was this prevented? Even without any quantitative statistical analysis, this could still create a sample bias in the data. If they were only permitted to take the survey one time, was it made clear with every question that they were providing answers for a very specific client set (their natural male competitors or PED-using females), even though they may have clients that fit multiple categories?
Results
Tables 1-3. I was able to follow the information in the tables and relate it to the text. I did think it would be easier for the reader if the information in the tables “lined up” better. Rather than organizing everything by tally (highest count on top), when data is given for the different subcategories of bodybuilders, perhaps align like-responses. There were some occasions where this could be done without abandoning the tally count order (moving “HIIT” to the bottom for PED-males or switching of “Athlete’s Preference” and “LISS+MISS+HIIT” for female natural competitors in table 1). Tables 2 (supplements) and 3 (PEDs) would require leaving the organization scheme used. For these, I suggest listing all the supplements/ drugs and having a column for each competitor category with the “Times Mentioned” and “Dosage range”. While it was not overtly difficult to follow as is, I do believe having alignment of the data in the tables would make male/female and natural/PED comparisons easier. Notably, comparisons between males/females or natural/PED competitors was not the focus of the paper, so the suggested alignment is a suggestion and not a necessary change. However, it would also allow a categorical arrangement of supplements/drugs in tables 2 & 3 (anabolics, fat burners, anti-inflammants, etc.), which could help clarify which types of these ergogenics are used most often, which could enhance the descriptive nature of the paper.
Discussion
In lines 183-191, it is suggested that individuals with greater muscle may need higher relative protein intake, and a paper [6] is cited. The problem of a lower-than-recommended intake in the control groups in the cited paper is presented. As such, the results from the paper are not valid, making this suggestion unfounded. Without evidence to substantiate such a claim, I suggest its removal.
Similarly, in lines 232-235, results from a single study are contrasted with conclusions reached from a meta-analysis. Clearly, much more weight should be placed on the review/meta. I would emphasize that acutely, fat use is enhanced during fasted exercise. Long term, there does not seem to be an advantage to exercising in the fasted state. However, these results come into question as many people simply take in excess calories to make up for a period of fasting. Given the strict diets employed in the bodybuilding community, similar results cannot be assumed.
I would suggest caution for the recommendation of research into fasted exercise + the use of fat burners (lines 240-242) from a safety standpoint. This is not really something that we should be recommending in general.
The wording in the last sentence in section 4.2 (lines 260-262) reads as though the athletes. And coaches are the experts on this matter. They should be turning to the literature for answers. I would rephrase this to indicate that even the most rigorous scientific research may not be able to answer this due to the impact of the named individual differences.
In lines 328-332, it was suggested that the use of GH was because coaches believe it may enhance fat loss. Perhaps this is true, and perhaps there is evidence from the qualitative responses supporting this supposition. However, as there is no analysis or extrapolation of the responses, there is no support for this is offered in the paper. It could as easily be determined that, and some might argue that it would be more likely that, use of GH while engaging in fasted CV exercise is done not to enhance fat loss, but to preserve lean mass (anti-catabolic effect), akin to protein intake prior to CV exercise.
The first sentence of the final paragraph of section 4.4 (lines 347-348) is a good discussion point. However, only a single study [37] is used to support this. Could either additional studies be cited, or this sentence be re-phrased to state “…aligned with a previous research survey in…”?
Discussion of PED regulation and prevalence and the lack of quality studies of these substances further down in this paragraph is relevant information to the study, however; it is not really related to the alignment of reported results from coaches versus athletes. Would it be possible to build out a paragraph expanding on that first sentence (where did these lists overlap; where might the discrepancy come from, etc.) and separate that information into another paragraph?
In the limitations section (4.5) the willingness of coaches to participate was brought up. This is, of course, an excellent point. Do you have any idea how many coaches had the opportunity to participate? This could help future researchers more accurately estimate how to achieve the numbers they need for a similarly designed study.
Final note.
While it is normally a goal of our field to offer recommendations/guidelines to athletes/trainers/coaches, the bodybuilding community presents a special case, specifically revolving around the common and even accepted use of PEDs. For one, as mentioned in the manuscript, quality studies of PED use do not exist and are unlikely to be done given the ethical concerns surrounding their design -even if the drugs used were made legal, allowable dosages for research would never approach those used by these competitors. Moreover, given the many serious health risks that come with their use and the fact that many of them are illegal, the only recommendation we could ever make would be not to take these drugs at all. Secondly, because this has been the stance of our field, yet as stated in the paper, PED use is generally accepted in the bodybuilding community as a whole (via the use of these PEDs by, not every individual but, many of the competitors and lack of regulation by the governing bodies). Thus, this community has continually ignored our recommendations/guidelines. I am unsure if it is worth the effort to design and execute studies to provide evidence-based recommendations to a group that seems so disinterested in what we have to say. I realize that is simply my opinion. I only wanted to share it because I believe this project was thoughtfully done and completed in the interest of serving others.
Author Response
Hello,
Thank you for the constructive feedback on our manuscript. We believe that your insightful comments have greatly improved the reading and interpretation of our data. We have addressed each suggestion by the reviewers below (our responses written in blue) and our track changes have been made in the manuscript as well.
REVIEWER 1:
Materials and Methods
Mostly clear with one minor and one more impactful question:
Minor: Regarding social media distribution, if only a few platforms were used, could they be listed? If many, I do not believe including a list would be worthwhile.
We have included a parenthetical list of the social media platforms used (Instagram, Twitter, Facebook)
More impactful: Was there any strategy employed to account for coaches of athletes fitting into multiple categories (male/female and/or natural/PED-using competitors)? Would/Could these coaches fill out the survey multiple times, say once answering regarding the training of the male clients and once regarding the training of their female clients? If not, how was this prevented? Even without any quantitative statistical analysis, this could still create a sample bias in the data. If they were only permitted to take the survey one time, was it made clear with every question that they were providing answers for a very specific client set (their natural male competitors or PED-using females), even though they may have clients that fit multiple categories?
Thank you for this question as we appreciate the concern that this could present in relation to creating a sample bias in the data. Our administration of the questionnaire accounted for this concern. Specifically, participant coaches were only permitted to take the survey once. Each question was specific to the target demographic/client so that participant coaches knew exactly which kind of athlete (male/female, enhanced/natural, off-season/preparation) to tailor their response towards. An example of a question asked was “How much protein do your male enhanced clients typically intake in the off-season? (g/lb) (Put ‘N/A’ if you do not coach these athletes)”. If a participant coach did not feel qualified to answer a question or did not coach that demographic, the survey instructions asked for them to skip that question or write “N/A”. Additionally, the survey was designed to only show a set of questions to coaches that gave a qualifying answer for it. For example, if a participant coach answered “No, I only coach natural athletes” to the question “Do you coach enhanced athletes using PEDs?” they would not be shown any questions relating to PED use. Answering “Yes, I coach only enhanced athletes” or “Yes, I coach both enhanced and natural athletes” would prompt the survey to display all questions relating to PED use.
Results
Tables 1-3. I was able to follow the information in the tables and relate it to the text. I did think it would be easier for the reader if the information in the tables “lined up” better. Rather than organizing everything by tally (highest count on top), when data is given for the different subcategories of bodybuilders, perhaps align like-responses. There were some occasions where this could be done without abandoning the tally count order (moving “HIIT” to the bottom for PED-males or switching of “Athlete’s Preference” and “LISS+MISS+HIIT” for female natural competitors in table 1). Tables 2 (supplements) and 3 (PEDs) would require leaving the organization scheme used. For these, I suggest listing all the supplements/ drugs and having a column for each competitor category with the “Times Mentioned” and “Dosage range”. While it was not overtly difficult to follow as is, I do believe having alignment of the data in the tables would make male/female and natural/PED comparisons easier. Notably, comparisons between males/females or natural/PED competitors was not the focus of the paper, so the suggested alignment is a suggestion and not a necessary change. However, it would also allow a categorical arrangement of supplements/drugs in tables 2 & 3 (anabolics, fat burners, anti-inflammants, etc.), which could help clarify which types of these ergogenics are used most often, which could enhance the descriptive nature of the paper.
Thank you for this suggestion. We originally considered the suggested organizational structure for Tables 1-3. However, the nature of Table 2 led us to using the current format. Table 2 is meant to list the Top 5 recommended supplements for each category (as that was the specific question asked to participant coaches that reflects the responses seen). Some categories had ties between responses, so more than 5 supplements are listed, while some categories did not have ties, so only 5 supplements are listed. If Tables 1 and 3 were reorganized to list supplements/PEDs by groups or even alphabetically, it would change the distinction of a tie in Table 2 and make it harder to visually see why some categories have 5 supplements listed and some have more. For the sake of consistency, we chose to list all supplements/PEDs in Tables 1-3 by rank order of most-to-least mentions.
To help make this clear, we have added language to the text that introduces Table 2 explaining the “Top 5” nature of the table and the inclusion of ties.
We agree that there would be value in grouping supplements/PEDs into categories such as “anabolics” or “fat burners.” However, given the multifaceted nature of some supplements/PEDs (e.g. multi-vitamins, caffeine) we cannot reasonably group them without making assumptions as to why participant coaches listed them.
Discussion
In lines 183-191, it is suggested that individuals with greater muscle may need higher relative protein intake, and a paper [6] is cited. The problem of a lower-than-recommended intake in the control groups in the cited paper is presented. As such, the results from the paper are not valid, making this suggestion unfounded. Without evidence to substantiate such a claim, I suggest its removal.
We have made the requested revision and removed the citation.
Similarly, in lines 232-235, results from a single study are contrasted with conclusions reached from a meta-analysis. Clearly, much more weight should be placed on the review/meta. I would emphasize that acutely, fat use is enhanced during fasted exercise. Long term, there does not seem to be an advantage to exercising in the fasted state. However, these results come into question as many people simply take in excess calories to make up for a period of fasting. Given the strict diets employed in the bodybuilding community, similar results cannot be assumed.
We have made the requested revision, removing reference #14.
I would suggest caution for the recommendation of research into fasted exercise + the use of fat burners (lines 240-242) from a safety standpoint. This is not really something that we should be recommending in general.
We agree with your sentiment about not recommending the practice of pairing fasted exercise with fat burners. We have revised this recommendation placing an emphasis on the need for future research (not the actual practice) on safety (primary objective) and efficacy (secondary objective). To the authors’ knowledge, there is no evidence to suggest that has directly investigated the safety of combining fat burners with fasted cardio. Given that the participant coaches responses indicate a prevalence of this choice, research investigating the safety of the practice would be beneficial to the health and well-being of athletes under the influence of their coaches.
The wording in the last sentence in section 4.2 (lines 260-262) reads as though the athletes. And coaches are the experts on this matter. They should be turning to the literature for answers. I would rephrase this to indicate that even the most rigorous scientific research may not be able to answer this due to the impact of the named individual differences.
We have rephrased this sentence, emphasizing that neither athletes nor researchers would be able to provide a direct answer to the question.
In lines 328-332, it was suggested that the use of GH was because coaches believe it may enhance fat loss. Perhaps this is true, and perhaps there is evidence from the qualitative responses supporting this supposition. However, as there is no analysis or extrapolation of the responses, there is no support for this is offered in the paper. It could as easily be determined that, and some might argue that it would be more likely that, use of GH while engaging in fasted CV exercise is done not to enhance fat loss, but to preserve lean mass (anti-catabolic effect), akin to protein intake prior to CV exercise.
We have removed all mention of a desire to enhance fat loss.
The first sentence of the final paragraph of section 4.4 (lines 347-348) is a good discussion point. However, only a single study [37] is used to support this. Could either additional studies be cited, or this sentence be re-phrased to state “…aligned with a previous research survey in…”?
We have rephrased these sentences to make it clear that only a single study [37] is used to support our point.
Discussion of PED regulation and prevalence and the lack of quality studies of these substances further down in this paragraph is relevant information to the study, however; it is not really related to the alignment of reported results from coaches versus athletes. Would it be possible to build out a paragraph expanding on that first sentence (where did these lists overlap; where might the discrepancy come from, etc.) and separate that information into another paragraph?
We have separated this section into two paragraphs focusing on the comparison of reference #37 and the present study, as well as PED regulation, prevalence, and research.
In the limitations section (4.5) the willingness of coaches to participate was brought up. This is, of course, an excellent point. Do you have any idea how many coaches had the opportunity to participate? This could help future researchers more accurately estimate how to achieve the numbers they need for a similarly designed study.
Due to the nature of the study advertisements via social media, we do not know how many coaches had the opportunity to participate, nor would we be able to reasonably estimate.
Final note.
While it is normally a goal of our field to offer recommendations/guidelines to athletes/trainers/coaches, the bodybuilding community presents a special case, specifically revolving around the common and even accepted use of PEDs. For one, as mentioned in the manuscript, quality studies of PED use do not exist and are unlikely to be done given the ethical concerns surrounding their design -even if the drugs used were made legal, allowable dosages for research would never approach those used by these competitors. Moreover, given the many serious health risks that come with their use and the fact that many of them are illegal, the only recommendation we could ever make would be not to take these drugs at all. Secondly, because this has been the stance of our field, yet as stated in the paper, PED use is generally accepted in the bodybuilding community as a whole (via the use of these PEDs by, not every individual but, many of the competitors and lack of regulation by the governing bodies). Thus, this community has continually ignored our recommendations/guidelines. I am unsure if it is worth the effort to design and execute studies to provide evidence-based recommendations to a group that seems so disinterested in what we have to say. I realize that is simply my opinion. I only wanted to share it because I believe this project was thoughtfully done and completed in the interest of serving others.
Thank you for your thoughtful contribution and providing your opinion regarding research on enhanced competitors. We agree that coaches and athletes may not abstain from utilizing PEDs despite the abundance of evidence regarding its unwanted side effects. At the same time, it may be beneficial to make an attempt to illuminate where all parties involved currently stand, even if we do not think the parties are likely to listen to potential additions to our pool of literature. It would be difficult to make any changes to the current landscape without knowing where current practices lie, thus this study aimed to bring awareness. We hear your concerns, and appreciate your valuable feedback.
Reviewer 2 Report
L17. Provide PED on first use of “performance enhancing drugs”.
Ls 97-98. How often did this happen? Please provide that information.
Table 3. Can your provide the dose information as well?
Ls 304-305. What was the range for testosterone intake in the present study. I suggest to provide to allow that comparison with the 400-600 mg intake in Ref 28.
L314. Is “arm skinfold thickness”
Author Response
Hello,
Thank you for the constructive feedback on our manuscript. We believe that your insightful comments have greatly improved the reading and interpretation of our data. We have addressed each suggestion by the reviewers below (our responses written in blue) and our track changes have been made in the manuscript as well.
L17. Provide PED on first use of “performance enhancing drugs”.
This has been added to line 17.
Ls 97-98. How often did this happen? Please provide that information.
We added this information to the text. This occurred an average of 3.6 times throughout participant coaches.
Table 3. Can your provide the dose information as well?
Participant coaches were not asked to list dosages when listing PED recommendations for enhanced athletes. In their free-responses, several coaches made mention of using a “low dose,” but we do not have any additional numbers to provide on this table from those questions. Table 3 is a general list of PED recommendations separated only by gender, and therefore listing dosages on this table would not be representative of the nuance used in dosing PEDs (i.e off-season/preparation dosages, history of PED use/tolerance, etc.). However, participant coaches were asked about total PED dosages that they would expect for a higher-level vs. lower-level enhanced athlete and that information is on lines 168-172 beneath Table 3.
Ls 304-305. What was the range for testosterone intake in the present study. I suggest to provide to allow that comparison with the 400-600 mg intake in Ref 28.
Participant coaches were not asked to list dosages when listing PED recommendations for enhanced athletes. Participant coaches in the present study gave a range of 300-4000mg/wk for total PED usage (not just testosterone alone) for male enhanced athletes, which is listed in lines 302-303. We have provided some revisions to this sentence to improve clarity.
L314. Is “arm skinfold thickness”
We have revised to “arm skinfold thickness”